# Brain Metastases from Lung Cancer: Is MET an Actionable Target?

**DOI:** 10.3390/cancers11030271

**Published:** 2019-02-26

**Authors:** Giulia M. Stella, Alessandra Corino, Giulia Berzero, Stefan Kolling, Andrea R. Filippi, Silvia Benvenuti

**Affiliations:** 1Department of Medical Sciences and Infectious Diseases, Unit of Respiratory System Diseases, IRCCS Fondazione Policlinico San Matteo, 27100 Pavia, Italy; corinoalessandra@gmail.com (A.C.); stefan.kolling01@universitadipavia.it (S.K.); 2IRCCS Istituto Casimiro Mondino, 27100 Pavia, Italy; giulia.berzero@mondino.it; 3Department of Medical Sciences and Infectious Diseases, Unit of Radiation Therapy, IRCCS Fondazione Policlinico San Matteo, 27100 Pavia, Italy; a.filippi@smatteo.pv.it; 4Department of Molecular Therapeutics and Exploratory Research, IRCCS Candiolo Cancer Institute, 10060 Candiolo, Italy; silvia.benvenuti@ircc.it

**Keywords:** Brain metastases, Invasive Growth, Lung Cancer, actionable target

## Abstract

The process of metastatic dissemination begins when malignant cells start to migrate and leave the primary mass. It is now known that neoplastic progression is associated with a combination of genetic and epigenetic events. Cancer is a genetic disease and this pathogenic concept is the basis for a new classification of tumours, based precisely on the presence of definite genetic lesions to which the clones are addicted. Regarding the scatter factor receptors MET and Recepteur d’Origin Nantais (RON), it is recognised that MET is an oncogene necessary for a narrow subset of tumours (MET-addicted) while it works as an adjuvant *metastogene* for many others. This notion highlights that the anti-MET therapy can be effective as the first line of intervention in only a few MET-addicted cases, while it is certainly more relevant to block MET in cases of advanced neoplasia that exploit the activation of the invasive growth program to promote dissemination in other body parts. Few data are instead related to the role played by RON, a receptor homologous to MET. We have already demonstrated an implication of MET and RON genes in brain metastases from lung cancer. On this basis, the aim of this work is to recapitulate and dissect the molecular basis of metastatic brain dissemination from lung cancer. The latter is among the big killers and frequently gives rise to brain metastases, most often discovered at diagnosis. Molecular mechanisms leading to tumour spread to the brain are mostly unknown and in turn these tragic cases are still lacking effective therapies. Based on previously published data from our group, we aim to summarise and analyse the pathogenic mechanisms leading to activation of the scatter factor receptor in brain metastatic lesions of lung primaries, from the point of view of replacing the currently used empirical treatment with a more targeted approach.

## 1. Introduction

Lung cancer is among the big killers and is frequently associated with brain metastases, most often discovered at the time of diagnosis. Lung carcinoma includes a series of different diseases which can roughly be divided into two groups based on clinical and histo-pathological features: non-small cell lung cancer (NSCLC), accounting for almost 80% of lung cancer diagnoses, and small cell lung cancer (SCLC), responsible for the remaining 20%. In 2015, the World Health Organization adopted the classification recently developed by the International Association for the Study of Lung Cancer, the American Thoracic Society, and the European Respiratory Society [1]. NSCLCs are further subclassified as adenocarcinoma (ADC), squamous cell carcinoma (SCC), neuroendocrine tumours, large-cell carcinoma, adeno-squamous carcinoma, salivary-gland type tumours, and other more undifferentiated tumours. Adenocarcinoma is currently the most frequent histologic type and accounts for almost half of all cases. For resected tumours, the novel classification introduces new entities, namely adenocarcinoma in situ (AIS) and minimally invasive adenocarcinoma (MIA) to designate non-mucinous ADC of ≤3 cm in size, with either purely lepidic growth or predominantly lepidic growth and ≤5 mm invasion, respectively. For invasive ADC, the new classification introduces histological subtyping based on the predominant pattern of neoplastic growth: lepidic (formerly non-mucinous bronchioloalveolar ADC), acinar, papillary, micropapillary, and solid. Four variants of invasive ADC are defined: invasive mucinous (formerly mucinous bronchioloalveolar adenocarcinoma), colloid, foetal, and enteric. On the other hand, three variants that were considered in the previous classification have been eliminated: mucinous cystadenocarcinoma, signet ring cell, and clear cell ADC [2]. This classification has implications for the strategic management of tissue, particularly for small biopsies and cytology samples, in order to maximise high-quality tissue available for molecular studies. More than 75% of all histological types are related to a habit of tobacco smoking and the association is strongest for SCLC and SCC [3]. Growing evidence points out that lung cancers arising in smokers and in never-smokers should be thought of as separate entities, since they exhibit distinct epidemiological, clinical, and bio-molecular features. Second-hand smoke exposure explains some deaths among non-smokers, but many deaths are unrelated to tobacco exposure. Occupational and environmental exposures, as well as genetic characteristics, have been identified as risk factors for the development of lung cancer in both smokers and never-smokers.

Patients are usually asymptomatic in the early stages of the disease. This is related to the sparse pain–fibre innervation of the lungs and the significant respiratory reserve that both lungs provide. The lack of symptoms is particularly true for lung cancers that originate in the periphery of the lungs. Approximately 5–10% of lung cancer patients are asymptomatic at presentation [4]. These cancers are often detected during evaluation for unrelated medical problems or on a chest radiograph performed for preoperative screenings. Most symptomatic lung cancer patients present with an already advanced disease. Tumours can metastasise via lymphatic and blood vessels. Mediastinal lymph-node invasion is among the first expressions of tumour progression and the presence of mediastinal invasion is one of the most crucial elements relevant in determining the optimal treatment strategy. Lung carcinomas have some preferential sites for distant metastases, among which is the brain. Secondary cerebral lesions are most often related to small-cell lung cancer (SCLC). In recent years, brain metastases are increasingly seen in those ADCs carrying mutations in the epidermal growth factor receptor gene (EGFR) and EML4/ALK rearrangements, whereas SCCs tend to locally invade the thoracic wall in many cases. The incidence of brain metastases in patients with lung cancer is approximately 25%, with only 5% surviving beyond the first year after diagnosis. They originate from cancer cells that have spread through the bloodstream and are associated with a poor prognosis of 4–5 months of median survival [5]. Due to an increase in intracranial pressure as well as the number, localisation, and rate of growth of brain metastases, they are associated with the onset of many clinical signs and symptoms, including headaches, sometimes with vomiting or nausea, and mental changes, such as increasing memory problems, seizures, and dizziness. Patients with known cancer and neurological symptoms should always undergo appropriate diagnostic tests which include either a CT scan or magnetic resonance imaging. In some cases, a biopsy is needed to reach diagnostic confirmation. Single or solitary brain metastases in patients with good systemic performance status should be strongly considered for surgical resection which both confirms the diagnosis and provides definitive treatment of the lesion. Patients with poor systemic performance status and/or multiple brain metastases are candidates for whole brain radiotherapy (RT). Whole brain RT could be efficacious in improving symptoms and quality of life; multiple radiosurgery is an option for oligometastatic brain disease [6,7,8]. Most often, steroids and symptom palliation are the only therapeutic opportunity [9]. 

The molecular mechanisms regulating metastatic spread to the brain are largely unknown [10,11]. Currently, no prognostic or predictive markers for the management of metastatic lung cancer exist; no therapies are available for the advanced stage of the disease defined by brain invasion, leading to a high mortality rate. As a result, the disease burden associated with lung cancer is among the highest of all cancer types. Thus, it is evident that metastasis formation in lung cancer is a multifaceted process. While many mechanisms and genes/proteins involved in the process have been identified, a breakthrough has not been achieved yet. Growing evidence suggests that metastasization follows from the inappropriate activation of a genetic programme termed ‘invasive growth’, a physiological process that occurs during embryonic development and post-natal organ regeneration, driven by the MET proto-oncogene [12]. Preliminary data from our (and other) groups suggest a role of MET-driven invasive growth in brain metastatization from lung cancer. In this work, we recapitulate and provide a deeper analysis of the role played by the invasive growth programme during cerebral dissemination of lung cancer, mainly focussing on the key data that unveil MET as a potentially novel actionable target. 

## 2. Invasive Growth and Metastatic Spread

Burgeoning evidence indicates that invasive growth is executed by stem and progenitor cells and is usurped by cancer stem cells. The MET proto-oncogene, which is expressed in both stem and cancer cells, is a key regulator of invasive growth. MET encodes for the receptor tyrosine kinase (TKR) for the hepatocyte growth factor (HGF) or scatter factor. It is located on chromosome 7, band q31, and is constituted by 21 exons separated by 20 introns codifying for a trans-membrane tyrosine kinase receptor synthesized as a single–chain precursor which undergoes post-translational cleavage into two disulfide-linked α and β subunits. The extracellular domain comprises two main regions, both involved in ligand binding. The first is known as SEMA domain, due to its homology with semaphorins, and includes the α chain and the N-terminal portion of the β chain. The second is the “immunoglobulin-like domain,” containing four disulfide-linked loop structures. The intracellular domains of MET include three functional portions: (i) the juxtamembrane sequence, including the Ser975 residue, which, upon phosphorylation, downregulates kinase activity; (ii) the catalytic region, containing Tyr1234 and Tyr1235 residues, which, upon receptor dimerization and transphosphorylation, upregulate kinase activity; (iii) the carboxy–terminal sequence. The residues Tyr1234 and Tyr1235, located in the catalytic domain, are critical for receptor activation. After that, MET elicits intramolecular phosphorylation of the other two critical tyrosine residues (Tyr 1349 and Tyr 1356) at the C-terminal of the α-chain. These two sites and the surrounding amino acids constitute the so-called “multifunctional docking site”, a motif that, when activated after phosphorylation, induces a series of biological processes that leads to invasive growth. In distinct cells and tissues, MET-driven specific activities are fulfilled by dedicated signaling cascades, with some transducers dominating over others according to the context, timing, and biological complexity [12] (Figure 1a). It acts as a sensor of adverse microenvironmental conditions (e.g., hypoxia and ionizing radiation) and drives cell invasion and metastasization through the transcriptional activation of the “invasive growth signature”, a genetic program leading to cell scattering, invasion, protection from apoptosis, and angiogenesis [3,13]. In human cancers, deregulated MET signaling can be achieved through: (i) genetic alterations: amplification (gastric carcinomas, liver metastases from colon cancer, lung), point mutations (kidney, hepatocellular, gastric, head and neck carcinomas, both in hereditary and sporadic carcinomas), establishment of autocrine loops (glioblastoma); (ii) overexpression consequent to transcriptional upregulation (in response to micro-environmental conditions such as hypoxia or ionizing radiation) (gastro-intestinal tract, thyroid, prostate, mammary carcinomas) [3]. The RON (Recepteur d’Origine Nantais) receptor—also known as the macrophage-stimulating receptor-1 (MSTR1)—belongs to the family of tyrosine kinase receptors of which MET is the prototype and displays 25% and 63% homology with its sibling receptor MET in the extracellular region, and 63% within the TK domain, respectively [14]. Through mechanisms analogous to MET, (Recepteur d’Origin Nantais) RON signaling results in invasive growth [15].

With respect to lung cancer, MET gene amplification occurs in about 4% of lung ADCs and 1% of SCC [16,17]. Accumulating preclinical and clinical evidence suggests that MET amplification behaves as an "oncogenic driver" and thus represents an actionable therapeutic target [18]. Notably, the emergence of MET-amplified clones has also been documented after treatment failure with tyrosine kinase inhibitors (TKI). Engelman et al. demonstrated the development of MET amplification in the HCC827 NSCLC cell line after exposure to increasing concentrations of the TKI gefitinib [19]. Cells lines that developed gefitinib resistance contained amplification of the MET-containing region 7q31.1 to 7q33.3. In the assessment of tumor tissue from 18 gefitinib-resistant NSCLC patients, 22% demonstrated MET amplification. Bean et al. also studied tissue from lung ADC patients in whom gefitinib or erlotinib resistance developed and found MET amplification in 21% of the cases. On the other hand, only 3% of patients who had not been treated with those drugs showed MET amplification [20]. Hence, amplification of the MET oncogene allows tumors to potentially overcome therapeutic inhibition of growth signals. Moreover, MET amplification has been reported to be associated with reduced progression-free survival and overall survival in EGFR-mutated cancers treated with the novel EGFR inhibitor osimertinib [21]. Thus, in a range from 15 to 20% of EGFR mutated cases, the selective pressure exerted by EGFR blockade through targeted therapies leads to the emergence of MET amplified subclones. The latter results in kinase overexpression and constitutive activation, ultimately imposing resistance to EGFR inactivation [22]. More recently, it has been reported that mutations in the MET exon 14 splice sites that cause exon 14 skipping are oncogenic. These genetic alterations are found in a relatively elderly population of patients with NSCLC, enriched in sarcomatoid histologies and accounting for 8–22% of cases, with an average prevalence of about 3% of ADCs and 1% of SCCs. Few data are available about the concomitant detection of MET gene amplification and exon 14 skipping, however, concurrent MET amplification has been reported in 15–21% of MET exon 14 positive NSCLC, and METY1003X mutations account for around 2% of the MET exon 14 alterations in NSCL [23,24,25]. These tumors can respond to MET-directed targeted inhibitors (e.g., rizotinib, cabozantinib, capmatinib, tepotinib, and glesatinib) [26,27,28,29]. The prognosis of patients with MET exon 14 skipping is reported similar to that of patients with major driver mutations [26]; several studies are already challenging the inevitable development of drug resistance [30,31,32]. 

## 3. MET-Driven Invasive Growth in Brain Metastases from NSCLC: A Proposed Hypothesis for a Still Obscure Phenomenon

The understanding of the molecular mechanisms underlying tumor spread to the brain is still in its infancy and, in turn, effective treatment for these tragic cases is yet to arrive. The concept that cancer mutated kinases molecularly mark druggable targets has led to intensive efforts to survey the kinome across a wide spectrum of human tumor types for mutations [33] and to the development of several targeted inhibitors [34]. It is well known that scatter factor receptors are oncogenes necessary for a limited subset of tumors (hence, also being called addicted), while they work as adjuvant *metastogenes* for many others. This notion highlights that anti-MET and anti-RON therapy can be effective as the first line of intervention in aforementioned addicted cases, whereas it is certainly more relevant to block MET and RON in cases of advanced neoplasms that exploit the activation of the invasive growth program to promote distant dissemination (Figure 2) On this basis, our work hypothesis has been focused on the analysis of the activation status of scatter factor receptors in brain metastases from NSCLC. We already assessed the whole MET and RON mutational profile in two relevant series of surgical samples of lung cancer and analyzed both the primary lung cancers and the lung-cancer derived brain lesions. The somatic origin of each mutation found was confirmed by sequencing the matched normal DNA. Mutations were detected only in malignant tissues. Overall, we found that [35,36]: (i) MET is mutated at a high frequency in brain metastases from NSCLCs (7.4%) compared with primary NSCLCs (4.4%) or an unselected cancer population (1–6%, data from COSMIC database, website at https://cancer.sanger.ac.uk/cosmic). Notably, the mortality rate after brain radiotherapy was significantly higher in tumors carrying somatic MET mutations compared with euploid wild-type MET lesions (*p* < 0.008) (Figure 1b); (ii) RON is mutated at a high frequency in brain metastases from lung cancers (9.5%) compared with an unselected cancer population (1%, data from COSMIC database, website at https://cancer.sanger.ac.uk/cosmic). In silico analysis suggested a damaging role of the changes found.

Moreover, the vast majority of MET mutations found in metastatic lesions affected the extracellular SEMA domain of the receptor, with the *E168D* change being the most frequent one. As discussed above, the SEMA domain of MET, which is shared with semaphorins, plexins, as well as the RON receptor, consists of a highly conserved variant form of the seven-blade β-propeller fold, defined by a set of cysteine residues, which form four disulphide bonds to stabilize the structure [37]. While the role of the intracellular MET tyrosine kinase domain has been fully investigated, the extracellular domain of MET is still poorly characterized. The non-catalytic SEMA domain is necessary for dimerization in addition to HGF binding [38] and has also been found to be involved in neoplastic invasiveness by biochemical characterization of SEMA mutants, albeit no clear mechanistic explanation is given [39]. Indeed, in vitro studies showed that SEMA mutated cells featured an increased proliferation rate, motile phenotype, invasion capacity, and even anchorage-independent growth capacity. Notably, this oncogenic potential was quite unexpected, since mutations do not affect receptor phosphorylation. 

Although preliminary, these study results do not disprove the initial premise, since it is conceivable that changes affecting the SEMA domain sequence may be reflected in structural alterations of the extracellular portion of the receptor. This could, in turn, affect the physical interactions between metastatic cells and the surrounding microenvironment, thus facilitating their highly invasive properties. Contrary to expectations, preliminary data pointed out that cell plasticity characterizing SEMA-mutated clones does not rely on increased deformability and adaptation to foreign environments. The motile behavior of SEMA-mutated cells seems to be unrelated to matrix stiffness. This finding was quite surprising, as a high migration ability, as well as an increased elasticity and deformability, were hypothesized to be involved in dissemination of metastatic cells to the brain. Overall, the SEMA mutants displayed: (i) an altered Young’s module (tension/deformation ratio); (ii) activation of interatomic interactions between cell surface and surrounding stroma when the cell was deformed; (iii) an altered extracellular matrix (ECM)/cell stiffness balance. It is well known that morphogenesis and metastases seem to arise from the same genetic program that instructs cells to undergo a biological process named *anoikis*. Through a mechanism known as epithelial-mesenchymal transition (EMT), cancer cells acquire a metastatic phenotype [40]. Metastatic cells reach a secondary site via blood or lymphatic vessels; after extravasation and the arrest of tumor cells in distant organs, the EMT process could be reverted through a mesenchymal-epithelial transition (MET); this last step coincides with cell repolarization and terminal differentiation in a tissue pattern that usually resembles branching tubules [41]. Cancer cells can increase their traction forces in response to increased ECM stiffness, documented by their phenotype transition, and they must also degrade ECM. Based on our findings, we presented the hypothesis that a shift in the biomechanics plays a crucial role, centered on the concept that the forces of deformation push cell dissemination [42]. Our preliminary data indicated that the plasticity exhibited by SEMA-mutated cell clones is not relying on increased deformability and adaptation to foreign environments. Instead, mutations affecting the extracellular SEMA domain might result in an unexpected interaction between mutated cells and the ECM, which in turn favors cellular scattering.

This crucial point may open new avenues for research focused on a deeper understanding of tumor progression with relevant diagnostic and therapeutic implications. Overall, the rationale of targeting MET in NSCLC has a controversial history. MET activation is responsible for about 20% of cases of resistance to EGFR inhibitors, and there is increasing evidence regarding the sensitivity to anti-MET inhibitors in the absence of concurrent EGFR mutations or MET ex 14 skipping variants [4]. To the best of our knowledge, the above data are the first reports of scatter factor activation in brain lesions from lung cancers. Limitations of the present report are related to the retrospective design of the study and the absence of a mechanistic explanation underpinning our findings. Consequently, the generated hypothesis is that the found MET and RON mutations might have clinical and prognostic implications as: (i) functional markers of highly aggressive lung tumors; (ii) actionable targets for a personalized approach in patients suffering from metastatic dissemination. Moreover, we reported an increased frequency of MET and RON mutations in a series of brain metastases from NSCLC and enhanced radio resistance in MET-mutated lesions [5,6]. Together, these data sustain a strong rationale for a deeper investigation of the invasive growth activation in metastatic lung cancer and highlight the need for identification of novel genes and key oncogenic pathways involved in NSCLC and SCLC progression.

## 4. Therapeutic MET Targeting in Brain Metastases

To successfully target brain metastatic lesions, therapeutic molecules have to be able to pass across the blood–brain barrier (BBB), which exists between the blood microcirculation system and the brain parenchyma. Its integrity is relevant in protecting the brain from systemic toxins, in assuring adequate nutrient levels, as well as in maintaining local homeostasis. As a net effect, it renders the brain an anatomical sanctuary site. It is anatomically and functionally distinct from the blood–cerebrospinal fluid barrier and the choroid plexus [43]. Blood flow alterations and altered vessel permeability are considered key determinants in the pathophysiology of brain injuries. Many signaling factors are known to control BBB permeability, including growth factors, miRNAs, and matrix metalloproteinases [44]. The availability of drugs to cross the BBB depends, among other factors, on their size [45]: The intact BBB is impenetrable to large macromolecules, including antibody-based proteins, although several approaches to increase drug delivery to brain tumors are currently under investigation. Small molecules account for the vast majority of available central nervous system drugs, primarily due to their ability to penetrate the phospholipid membrane of the BBB by passive or carrier-mediated mechanisms [46]. However, brain metastases appear to be resistant to most conventional systemic anticancer treatments. The seed and soil concept might take into consideration both the intrinsic properties of tumor cells and their capacity to destroy the BBB [47]. In breast cancer patients, brain metastatic cells expressing high levels of activated MET promote the metastatic process via upregulation of inflammatory cytokines and vascular reprogramming [48]. Pterostilbene (PTER) is a potential agent to treat brain metastases by targeting said MET-mediated perivascular growth and angiogenesis. 

In Section 3, we described how the occurrence of MET mutations in lung cancer cells—mainly affecting the SEMA domain of the receptor—activates their invasive potential with a preferential tropism to the brain. Thus, based on these preliminary results, mutated MET is emerging as a novel actionable target for these difficult-to-treat cases. Many MET inhibitors have been developed and/or have entered clinical evaluation in recent years. A detailed description of each molecule is beyond the scope of this review, and we have limited our discussion to those drugs which appear most promising to target secondary brain lesions, due to their biochemical properties and structure. Scarce data is available on the role of monoclonal antibodies inhibiting MET in controlling brain metastases. Onartuzumab does not improve the clinical outcome in primary brain tumors, such as glioblastoma [49]. An in vitro and in vivo screening showed Sym015, consisting of two humanized monoclonal antibodies directed against non-overlapping epitopes of MET, to be more effective than emibetuzumab, a monoclonal IgG4 antibody against MET currently in clinical development, in inhibiting MET-amplified tumors (as a mechanism to overcome resistance to EGFR-targeting agents in advanced NSCLC) [50]. Data regarding the ABT-700 antibody show similar results [51]. The recently developed IgG2-enhanced next generation MET monoclonal antibody KTN0073 exhibits potent anti-tumor properties both in vitro and in vivo, not only on MET-amplified cells, but also in the juxta-membrane exon 14 deletion mutants [52]. The phage-derived anti-MET antibody 7A2/107_A07 competes with HGF, the endogenous MET ligand, as well as the HGF fragment NK1, by binding the IG1 domain of the receptor rather than the SEMA domain. In conclusion, no clear data are available about the role of monoclonal antibodies in halting MET-driven brain metastatic growth. Moreover, the “selective” occurrence of somatic mutations causes one to think that this subset of patients could benefit from small molecule inhibitors. Among the novel small inhibitors, cabozantinib, a MET, RET, and VEGFR2 inhibitor, has been reported to be effective in radioresistant MET-mutated brain metastases from renal cell carcinoma [53] and to show rapid intracranial response in crizotinib-resistant MET-exon14 positive NSCLC [54]. On the other hand, the concomitant activation of the MET receptor and the ALK fusion gene has been reported to be associated with a rapid response to crizotinib in NSCLC brain metastatic lesions [55]. To date, no data are available on the potential efficacy of the novel small inhibitor glesatinib, although it shows promise in overcoming resistance to type I anti-MET agents [56]. The combination of osimertinib and the MET inhibitor savolitinib showed enhanced efficacy for pre-treated patients with MET-positive, EGFR-mutant NSCLC, regardless of prior treatment with a T790M-directed therapy, with the T790M mutation being frequently detected in brain metastases of lung tumors (the TATTON trial, website: www.ClinicalTrials.gov, identifier: NCT02143466). Capmatinib, an orally available highly potent and selective inhibitor of MET, demonstrated a manageable toxicity profile in treatment–naïve patients with NSCLC and MET exon14 mutation. Preliminary data reported that capmatinib is able to pass the BBB and it is active in brain. In in vivo models, the combination of capmatinib and the pan-EGFR inhibitor afatinib completely suppressed tumor growth in mice orthotopically injected with cells derived from brain metastasis from NSCLC patients [57]. All in all, the EGFR-MET crosstalk is critical for aggressive behavior of NSCLCs and occurs through activation of MAP kinases [58] and, hence, pharmacological inhibition of MAP kinases (such as by the MEK inhibitor selumetinib) can enhance MET signaling [59]. Altiratinib (DCC-2701) was instead designed based on the rationale of engineering a single therapeutic agent able to address multiple hallmarks of cancer, among which was MET. This agent exhibits properties amenable to oral administration and exhibits substantial BBB penetration, an attribute of significance for potential treatment of MET-positive metastatic clones [60]. More recently, the novel PLB-1001 compound showed a better BBB penetrance than other MET inhibitors, with an acceptable safety profile and achieving partial responses in MET-mutated glioblastoma [61]. It is an orally administered compound which acts as potent, highly selective competitor for ATP binding. This preliminary data opened the way to further evaluation of PLB-1001’s efficacy in treating secondary brain lesions from different primary sites. Notably, quite recent data underline that HGF/MET signaling is involved in the immune response, mainly in modulating dendritic cell function. MET inhibition in animal models promotes adoptive T-cell transfer and boosts check-point immunotherapy by increasing T cell infiltration in tumors, independently of the status of MET addiction of neoplastic cells. This finding suggests the therapeutic opportunity of co-treatment in advanced cancers [62,63,64,65]. Radiotherapy is the most widely used therapeutic approach in brain metastases; however, in some cases resistance to ionizing radiation is responsible for therapeutic failure. MET activation is implicated in inducing tumor radio-resistance and the occurrence of MET mutations in brain lesions from NSCLC has been associated with a higher mortality after RT. Hence, combining radiotherapy with MET targeting is critical to improve the treatment outcome of those lesions. Synergistic antitumor effects are well documented both in in vitro and in vivo models [66,67], although no data from clinical trials are yet available.

## 5. Conclusions

Over the past few years, the improved knowledge of the biological, genetic, and molecular heterogeneity of tumors, together with the development of improved pharmacological technologies, has allowed the identification of molecular targets for novel therapeutic strategies. This fast process has led to the overall reconsideration of the biological and genetic peculiarities that can make each tumor a pathology of its own. The identification of patients likely to respond to specific treatments according to the presence of relevant molecular targets (personalized medicine) needs clinical studies focused on a constant and productive interaction among the professionals with a significant background in the various disciplines. The process of metastatic dissemination begins when malignant cells start to migrate and leave the primary mass. Cancer is a genetic disease and this pathogenic concept is the basis for a new classification of tumors, based precisely on the presence of definite genetic lesions to which the clones are addicted. Growing evidence suggests that cancer cells inappropriately execute the MET-driven invasive growth program which is actively involved in tumor onset and progression. The presented data sheds new light on MET, and its sibling RON, as promising candidate genes in both onset and therapy of secondary brain metastases. Further mechanistic validation is needed to test the presented hypothesis. If experimentally confirmed, future therapies could take advantage of novel MET inhibitors specifically designed to cross the BBB and could possibly enhance the radio-sensitivity of brain lesions. 

## Figures and Tables

**Figure 1 cancers-11-00271-f001:**
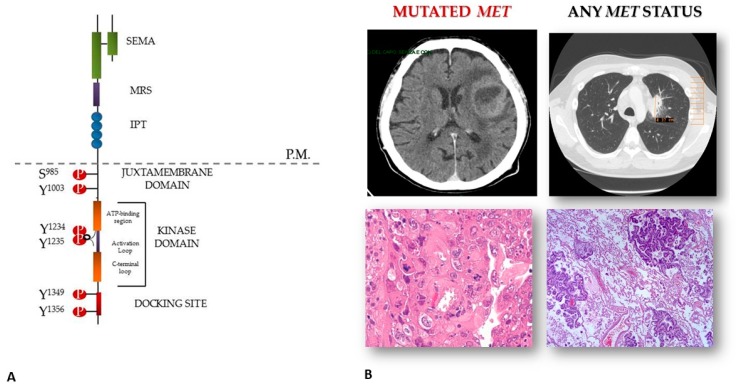
MET activation in brain lesions from non-small cell lung cancer (NSCLC). Panel (**A**) structure of the MET receptor; panel (**B**) brain metastases and the matched primary lung mass. The lesions feature similar histo-morphology, but different MET status, with MET mutated cells being only detected in the secondary lesion.

**Figure 2 cancers-11-00271-f002:**
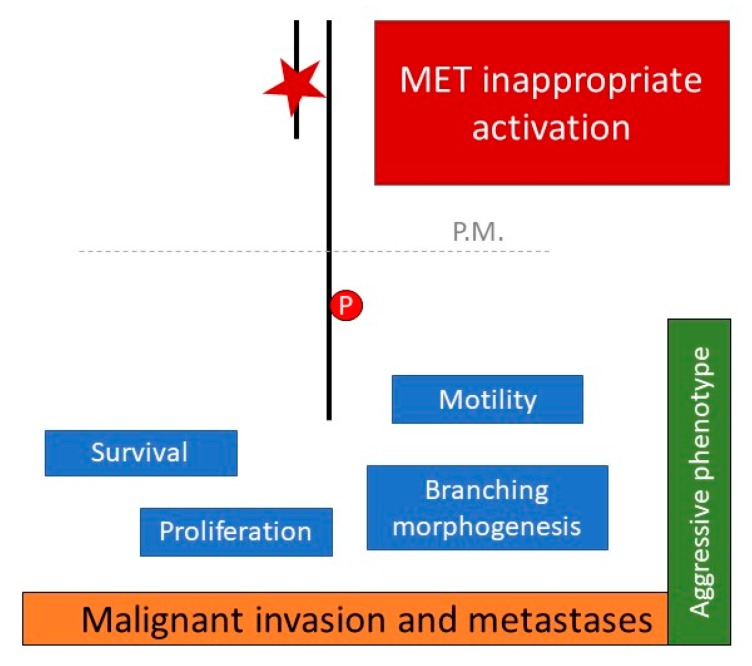
The MET-driven invasive growth. Biological features of the invasive growth program orchestrated by the MET oncogene. In the case of brain metastases from lung cancer, MET activation is related to the occurrence of somatic mutations (✶) affecting the receptor’s extracellular SEMA domain.

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
