# Peer review of "Brain Metastases from Lung Cancer: Is MET an Actionable Target?"

_cancers, 2019, doi:10.3390/cancers11030271_

Reviewer 1 Report

This paper focuses on a fascinating hypothesis. The authors assert that the scatter factor receptors MET and RON have a predominant role in lung cancer-related brain metastasis development.

The available data from literature are weak to sustain this hypothesis.

The paper need to be profoundly revised:

-          It appears too long with many non-informative parts

-          It should be rewritten as a hypothesis or proposal

-          The authors should explain better the currently known mechanism of MET alterations as an oncogene driver and as a mechanism of resistance to EGFR-TKIs.

-          The classification of lung cancer histology in the introduction section should be revised.

-          English language should be revised and some typing errors corrected.

Author Response

Response to Reviewer 1

This paper focuses on a fascinating hypothesis. The authors assert that the scatter factor receptors MET and RON have a predominant role in lung cancer-related brain metastasis development. The available data from literature are weak to sustain this hypothesis. The paper needs to be profoundly revised:

We would like to thank the reviewer for careful and thorough reading of this manuscript and for the thoughtful comments and constructive suggestions.

Comment 1

-          It appears too long with many non-informative parts

Response 1

We agree with this comment which help to improve the quality of the manuscript. Accordingly, we eliminate the non-informative parts in the text

Comment 2

-          It should be rewritten as a hypothesis or proposal

Response 2

The whole text has been revised as suggested. In detail we introduce the following statements:

Line 190:” MET-driven invasive growth in brain metastases from NSCLC: a proposed hypothesis for a still obscure phenomenon”

Line 201: “On this basis our work hypothesis has been focused on the analysis of activation status of scatter factor receptors in brain metastasis from NSCLC...”

Line 228: “Although preliminary, these results study do not disprove the initial premise…”

Line 262: “Limitations of the present report are related to the retrospective design of the study and the absence of a mechanistic explanation underpinning our findings. The consequently generated hypothesis is that the found MET and RON mutations might feature have clinical and prognostic implication…”

Line 301: “Thus, based on these preliminary results,  mutated…”

Line 367:“Further mechanistic validation is needed to test the presented hypothesis. If experimentally confirmed …”

Comment 3

-          The authors should explain better the currently known mechanism of MET alterations as an oncogene driver and as a mechanism of resistance to EGFR-TKIs.

Response 3

The text has been implemented according to Reviewer’s suggesting, as follows (from line 148:” Accumulating preclinical and clinical evidence suggests that MET amplification behaves as an "oncogenic driver" and thus represents an actionable therapeutic target [Kawakami H, Okamoto I, Okamoto W, Tanizaki J, Nakagawa K, Nishio K. Targeting MET Amplification as a New Oncogenic Driver. Cancers (Basel). 2014;6(3):1540-52. Published 2014 Jul 22. doi:10.3390/cancers6031540]. Notably, the emergence of MET-amplified clones has also been documented after treatment failure with tyrosine kinase inhibitors (TKI). Engelman et al. demonstrated the development of MET amplification in the HCC827 NSCLC cell line after exposure to increasing concentrations of the TKI gefitinib [Engelman JA, Zejnullahu K, Mitsudomi T et al. MET amplification leads to gefitinib resistance in lung cancer by activating ERBB3 signalling. Science. 2007; 316(5827):1039-43]. Cells lines that developed gefitinib resistance contained amplification of the MET-containing region 7q31.1 to 7q33.3. In the assessment of tumour tissue from 18 gefitinib-resistant NSCLC patients, 22% demonstrated MET amplification. Bean et al. also studied tissue from lung ADC patients in whom gefitinib or erlotinib resistance developed and found MET amplification in 21% of cases. On the other hand, only 3% of patients who had not been treated with those drugs showed MET amplification [Bean J, Brennan C, Shih JY et al. MET amplification occurs with or without T790M mutations in EGFR-mutant lung tumours with acquired resistance to gefitinib or erlotinib. Proc. Natl. Acad. Sci USA. 2007, 104, 20932-20937]. Hence, amplification of the MET oncogene allows tumours to potentially overcome therapeutic inhibition of growth signals. Moreover, MET amplification has been reported to be associated with reduced progression-free survival and overall survival in EGFR-mutated cancers treated with the novel EGFR inhibitor osimertinib [Wang Y, Li L, Han R, Jiao L, Zheng J, He Y.  Clinical analysis by next-generation sequencing for NSCLC patients with MET amplification resistant to osimertinib. Lung Cancer. 2018 Apr;118:105-110]”. 

Comment 4

-          The classification of lung cancer histology in the introduction section should be revised.

Response 4

We thank the Reviewer for this careful analysis. The text has been improved as follows (from line 40: “…the World Health Organization has adopted the classification recently developed by the International Association for the Study of Lung Cancer, the American Thoracic Society, and the European Respiratory Society [Travis WD, Brambilla E, Nicholson Ag et al. The 2015 World Health Organization Classification of Lung Tumors. J Thor Oncol 2015; 10(9):1243-60]. NSCLCs are further subclassified as adenocarcinoma (ADC), squamous cell carcinoma (SCC), neuroendocrine tumours, large-cell carcinoma, adeno-squamous carcinoma, salivary-gland type tumours and other more undifferentiated tumours. Adenocarcinoma is currently the most frequent histologic type and accounts for almost half of all cases. For resected tumours, the novel classification introduces new entities, namely adenocarcinoma in situ (AIS) and minimally invasive adenocarcinoma (MIA) to designate non-mucinous ADC of ≤ 3 cm in size, with either purely lepidic growth or predominantly lepidic growth and ≤ 5 mm invasion, respectively. For invasive ADC, the new classification introduces histological subtyping based on the predominant pattern of neoplastic growth: lepidic (formerly non-mucinous bronchioloalveolar ADC), acinar, papillary, micropapillary, and solid. Four variants of invasive ADC are defined: invasive mucinous (formerly mucinous bronchioloalveolar adenocarcinoma), colloid, foetal, and enteric. On the other hand, three variants that were considered in the previous classification have been eliminated: mucinous cystadenocarcinoma, signet ring cell, and clear cell ADC [Truni A, Santos Pereira A, Cavazza A et al. Classification of different patterns of pulmonary adenocarcinomas. Expert Rev Respir Med. 2015 Oct;9(5):571-86”

Comment 5

-          English language should be revised (by Dr Stefan Kolling , who has been added in the author list) and some typing errors corrected.

Response 5

The text has been improved in Its English language and typing errors have been corrected.

Reviewer 2 Report

This paper is an excellent review of the role of MET in metastasis process especially in the case of brain metastasis of lung carcinomas with numerous data on the mechanistic implications of MET in tumor invasion and metastasis.

My only remark is that the authors focus their studies on the altered extracellular matrix/ cell stiffness balance associated with MET mutations. The relation of epithelial to mesenchymal transition is not evoked in their article.It would be nice to get some information on that.

Author Response

Response to Reviewer 2

Comment 1

This paper is an excellent review of the role of MET in metastasis process especially in the case of brain metastasis of lung carcinomas with numerous data on the mechanistic implications of MET in tumor invasion and metastasis. My only remark is that the authors focus their studies on the altered extracellular matrix/ cell stiffness balance associated with MET mutations. The relation of epithelial to mesenchymal transition is not evoked in their article .It would be nice to get some information on that.

Response 1

We appreciate the positive feedback from the reviewer. The text has been implemented as follows (from line 239): “. It is well known that morphogenesis and metastases seem to arise from the same genetic programme that instructs cells to undergo a biological process named anoikis. Through a mechanism known as epithelial-mesenchymal transition (E.M.T.), cancer cells acquire a metastatic phenotype [Thiery JP. Epithelial-mesenchymal transition in tumor progression. Nat Rev Cancer 2002, 2:442- 454]. Metastatic cells reach a secondary site via blood or lymphatic vessels; after extravasation and the arrest of tumour cells in distant organs, the EMT process could be reverted through a mesenchymal-epithelial transition (M.E.T.); this last step coincides with cell repolarisation and terminal differentiation in a tissue pattern that usually resembles branching tubules [Comoglio PM, Boccaccio C. Scatter factors and invasive growth. Semin Cancer Biol 2001, 11:153-165]. Cancer cells can increase their traction forces in response to increased ECM stiffness documented by their phenotype transition and they must also degrade ECM. Based on our findings, we presented the hypothesis that a shift in the biomechanics plays a crucial role, centred on the concept that the forces of deformation push cell dissemination [36]. Our preliminary data indicate that the plasticity exhibited by SEMA-mutated cell clones is not relying on increased deformability and adaptation to foreign environments. Instead, mutations affecting the extracellular SEMA domain might result in an unexpected interaction between mutated cells and the ECM which in turn favours cellular scattering.

Round  2

Reviewer 1 Report

I have reviewed the revised manuscript, and I think it is worthwile to be published in the present form, suggesting  an interesting connection between Met  and Ron oncogenes mutations and brain metastases in lung cancer. Although  the paper should be considered as Hypothesis generating  it could be considered  a very well written review  and can stimulate furter research in this area.